# Comparison of regional with general anaesthesia on postoperative delirium (RAGA-delirium) in the older patients undergoing hip fracture surgery: study protocol for a multicentre randomised controlled trial

Ting Li,[1] Joyce Yeung,[2,3] Jun Li,[1] Yan Zhang,[4] Teresa Melody,[3] Ye Gao,[1] Yi Wang,[1] Qianquan Lian,[1] Fang Gao,[1,2] on behalf of the RAGA-Delirium Investigators

For numbered affiliations see end of article.

**Correspondence to**
Professor Fang Gao;
f.g.smith@bham.ac.uk

## ABSTRACT

**Introduction** Postoperative delirium (POD) is a common serious postoperative complication especially in older people and is associated with increased mortality, morbidity and healthcare costs. There is no clear consensus which anaesthesia is associated with less incidence of POD for older patients. We aim to assess whether regional anaesthesia results in lower incidence of POD comparing with general anaesthesia (GA) among older patients undergoing hip fracture surgery.

**Methods and analysis** RAGA-delirium is a pragmatic, multicentre, prospective, parallel grouped, randomised controlled clinical trial comparing RA or GA for hip fracture surgery. A total of 1000 patients who are 65 years or over and who are having planned hip fracture surgery in nine clinical trial centres of China will be randomised in a 1:1 ratio to receive either anaesthesia for the surgery. The primary endpoint will be the incidence of POD at day 7. The secondary endpoints will be the subtype, severity and duration of delirium, postoperative acute pain score, incidence of other postoperative non-delirium complications, quality of life and cost-effective outcomes. Randomisation will be performed at the patient level using computer-generated assignment. Outcome assessors will be blinded from intervention assignment. Assessments will be conducted before surgery, intraoperatively, postoperatively, during the hospital stay, at 30-day, 6-month and 1-year postoperative intervals.

**Potential impact of study** This study will provide clinical evidence with a more robust methodology to help anaesthetists in selecting appropriate anaesthesia for older patients with high risk for POD. At the era of increasing emphasis on delirium prevention, this trial has the potential to inform the future national guideline to reduce POD.

**Ethics and dissemination** Ethical approved by the local institutional review board. Trial results will be presented at national and international academic conferences, and published in peer-reviewed journals.

**Trial registration number** ClinicalTrials.gov (NCT02213380); pre-results.

## Strengths and limitations of this study

► This will be a randomised controlled trial with a more robust methodology to improve trial accuracy and decrease risk of potential bias.
► Methodology strengths of this trial include internet central randomisation, computer-generated assignment, blinded assessment and concealment allocation, and appropriate sample size estimation.
► The other strength is the pragmatic and multicentre design of nine study sites in order to depict the outcomes of anaesthesia methods in real-life routine clinical practice.
► The pragmatic and multicentre design may limit the consistency of interventions. However, this compromise of sample and measurement homogeneity is expected to be diluted with large sample size.

## BACKGROUND

Postoperative delirium (POD) is a common postoperative complication, especially in the elderly, and is associated with increased mortality, post-traumatic stress disorder, longer length of hospital stay, extra nursing requirements and increased healthcare cost.[1 2] Systematic reviews have shown a high incidence of delirium in surgery after hip fracture (4%–53%).[3] Currently there is no robust evidence demonstrating the efficacy of any specific treatment for POD, creating the urgent need for researches to investigate delirium prevention. The aetiology of delirium remains poorly understood, but studies have highlighted some pre-existing and precipitating factors such as anaesthetic agents that may be associated with predispose to POD.[4]

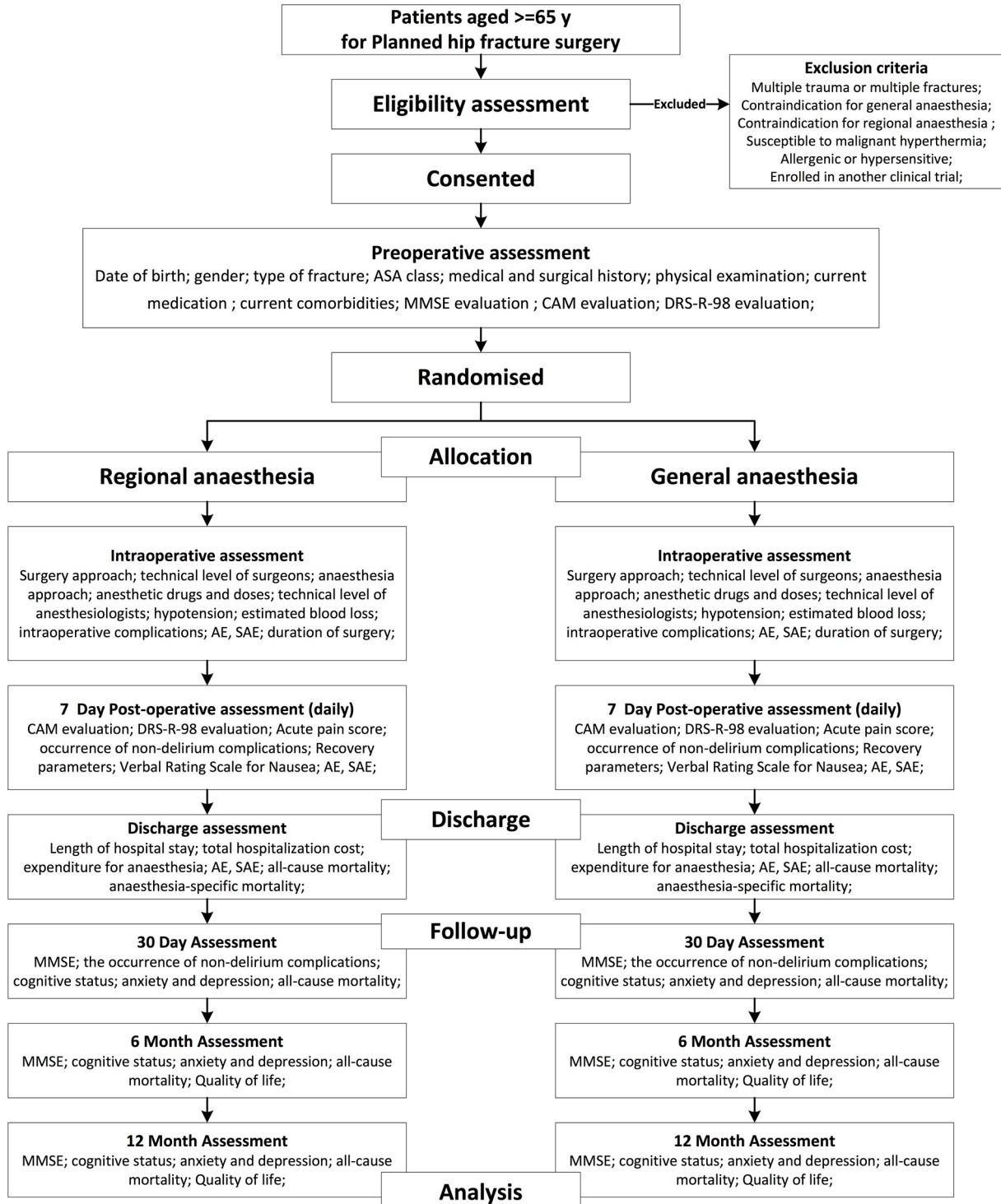

**Figure 1** Flow diagram of the RAGA trial. AE, adverse event; ASA, American Society of Anesthesiologists; CAM, Confusion Assessment Method; DRS-R-98, Delirium Rating Scale-Revised-98; MMSE, Mini-Mental State Examination; SAE, serious adverse event.

Anaesthesia may be classified into regional anaesthesia (RA) or general anaesthesia (GA). GA involves inducing sleep or loss of consciousness and is achieved by either inhalational agents or intravenous anaesthetic agents. RA involves an injection of local anaesthetic inside the spine (neuroaxial block) or around the nerves (peripheral nerve block) to prevent pain in the leg with the hip

fracture. Older patients with hip fracture can be frail and often have many multiple medical comorbidities associated with advanced age[5], which places them at high risk of morbidity and mortality after anaesthesia. There is no up-to-date prospective data[4] and clear consensus as to whether which anaesthesia can lead to better patient outcomes undergoing a surgery. At present both RA and

GA are administered during surgery for elderly people. However, the eventual choice of anaesthetic regimen used is based on the preference and experience of the anaesthetist alongside discussion with the patient and their carers rather than evidence-based decision-making.[6] Recently, a Cochrane systematic review based on six studies including a total of 624 participants claimed no difference in the risk of acute confusional state was found between different anaesthetic regimens. However, current reviews were based on old and low-quality evidence, such as poor allocation concealment, assessor blinding or small sample sizes;[7] thus any estimate of effect still remain uncertain.

Over the last 20 years, the elderly population is growing fast in most parts of the world[8 9] and the number of older people undergoing surgical procedures has increased.[10 11] Hip fractures are a global health problem, with over 1.6 million patients suffering hip fractures worldwide each year,[12 13] of which 680 000 occur among the 70 million elderly people in China.[14–17] 50% of total global hip fractures will occur in Asia by the year 2050.[18] The majority of people with hip fracture are elderly and are treated surgically, which requires anaesthesia.[19] However, no studies have yet investigated the effect of RA and GA on the POD in elderly patients undergoing hip fracture surgery in China. The selection of an optimal anaesthesia regimen that can achieve the ideal anaesthetic effect during operation while reducing the influence on postoperative brain function in elderly patients is a challenge for anaesthetists and calls for a high-quality multicentre clinical trial.

## METHODS AND DESIGN
### Study design
This is a pragmatic, multicentre, prospective, parallel grouped, randomised controlled clinical trial with cost-effective analysis. This follows the SPIRIT 2013 (Standard Protocol Items: Recommendations for Interventional Trials) statement.[20 21] Figure 1 shows the design of the study.

### Objectives
#### Primary objective
The aim of the RAGA trial is to determine whether different types of anaesthesia (RA vs GA) given to older patients undergoing hip fracture surgery result in equivalent incidence of POD.

#### Secondary objectives
► To compare the severity, type and duration of POD between the anaesthesia groups;
► To assess whether there are differences in incidence of POD in different patients (by age, dementia or delirious patients) between groups;
► To describe the frequency of analgesics, blood pressure, pain in the postoperative period after either RA or GA and explore other possible factors associated with increased risk of POD;

► To compare the occurrence of non-delirium complications between the groups;
► To assess and compare cost-effectiveness of using RA or GA in terms of medical costs;
► To follow-up and analyse long-term outcomes and time to event data such as long-term cognitive dysfunction, quality of life and other mortality, and morbidity indices after POD during hospital stay and 12 months after hospital admission.

### Participants
#### Study setting
This study will take place at nine hospitals in different regions of Mainland China. The hospitals have an annual total of 285 000 annual surgeries, of which 4580 were planned surgery for fragility hip fracture, with 2390 patients over 65 years old per year.

The nine investigational centres are the following: Department of Anaesthesiology, The Second Affiliated Hospital and Yuying Children's Hospital of Wenzhou Medical University, Wenzhou, China; Department of Anaesthesiology, Tongji Hospital, Tongji Medical college, Huazhong University of Science and Technology, Wuhan, China; Department of Anaesthesiology, The Central Hospital of Lishui City, Lishui, China; Department of Anaesthesiology, Lishui People's Hospital, Lishui, China; Department of Anaesthesiology, The Second Hospital of Ningbo, Ningbo, China; Department of Anaesthesiology, The sixth Hospital of Ningbo, Ningbo, China; Department of Anaesthesiology, The First Affiliated Hospital of Nanchang University, Nanchang, China; Department of Anaesthesiology, Taizhou Enze Hospital, Taizhou, China; Department of Anaesthesiology, The First Affiliated Hospital of Wenzhou Medical University, Wenzhou, China.

#### Study population
##### Inclusion criteria
► Patients ≥65 years old;
► Hospital admission for surgery for fragility hip fracture (fragility hip fractures are defined as femoral neck, femoral head, intertrochanteric or the subtrochanteric fractures);
► American Society of Anesthesiologists class I–IV;
► The ability to receive written informed consent from the patient or patient's legal representative.

##### Exclusion criteria
Patients presenting with one of the following criteria will not be included in the trial:
► Multiple trauma, multiple fractures or other fractures outside the inclusion criteria, such as pathological fractures, pelvic fractures, femur fractures;
► Contraindication for GA (drug allergies to GA or any other drugs administered during this trial);
► Contraindication for RA (infection at the site of needle insertion, coagulopathy, international normalised ratio >1.4, platelet count <80×$10^9$/L, allergy to local anaesthetics);

► Susceptibility to malignant hyperthermia;
► Current enrolment in a clinical trial of investigational medicine.

## Outcomes and measurements
### Primary outcome
► The primary outcome is the incidence of POD during 7 postoperative days, diagnosed with Confusion Assessment Method (CAM).[22]

### Secondary outcomes
► The subtype, severity and duration of delirium in 7 days after the surgery, diagnosed with the Delirium Rating Scale-Revised-98 (DRS-R-98)[23] and CAM;
► Duration of delirium, measured by days from the onset of delirium symptom to resolution of symptoms;
► The intensity of postoperative pain, measured with acute pain score in 7 days after the surgery using Visual Analogue Scale (VAS)[24];
► The occurrence of non-delirium in-hospital complications within 30 days after surgery, including chest infection, myocardial infarction, renal failure, gastrointestinal ileus;
► Length of hospital stay, measured by the sum of inpatient days from admission to discharge;
► Mortality during hospital stay and during follow-up period after hospital admission;
► Incidence of delirium in clinic or at residence on 30 day, 6 month, 12 month after surgery, diagnosed with CAM;
► Cognitive and functional decline on 30 day, 6 month, 12 month after surgery, diagnosed with Mini-Mental State Examination (MMSE) and Hospital Anxiety and Depression Scale;
► Quality of life on 30 day, 6 month, 12 month after surgery, using 36-Item Short Form questionnaire;
► Economic parameters including total cost in hospital and expenditure for anaesthesia and resource using costs, including equipment and disposables required for each anaesthetic technique.

## Assignment of interventions
### Randomisation
Randomisation will be performed at the patient level. Consecutively screened and eligible patients will be included in the trial at each centre after initiation of the study. Patients will be allocated in a 1:1 ratio into either RA group or GA group 1 day before surgery using computer-generated assignment (web or telephone developed by the study data management provider). Stratification will be performed to minimise group imbalances on these variables by the following factors: age (65–79, ≥80 years), presence of preoperative delirium (yes, no), perioperative dementia (yes, no). The investigator or designee will complete a randomisation worksheet as detailed in the study manual.

### Blinding
Owing to the nature of the study, it is not possible for the surgeon and anaesthetist delivering the interventions to be blinded to the treatment options.[25] In order to minimise any potential bias, investigators have been divided into two different teams: (1) Unblinded: the anaesthetist and other surgical staff who work in the operation theatre will know whether the participants have GA or RA. These staff will not be involved in the assessment of outcome measures. (2) Blinded: principal investigators (PIs), coinvestigators and statistician will not know patient's group allocation. Data collectors or outcome assessors, for example, the medical staff who provide postoperative care in the ward and visit patients for preoperational assessment, hospital visits and subsequent follow-ups, will be blinded from group allocation throughout the study.

## Implementation and data collection
### Treatment and comparator
There will be two arms of research: RA group and GA group. Conducting of interventions will follow the national or local routine clinical practice in China. For an overview of the schedule see figure 1.

Due to the nature of a pragmatic trial with only a few exclusion criteria, no further efforts are proposed to strictly standardise the applied interventions. In both groups, patients will receive anaesthesia according to local practice. The type of medication, dosage and additional pain medication will be based on clinical protocols of each study centre and be documented in detail in the medical record and in the Case Record Forms (CRFs). Experienced and qualified anaesthetists will be designated to perform the RA or GA. Prior to the study, study personnel are trained to follow the study protocol in accordance with the Good Clinical Practice (GCP) principles.

### Recruitment and grouping
Patients will be recruited through the department of orthopaedics. All possible candidate patients for the trial will be provided with information sheets and informed about the aim of the trial, the treatment allocations, the workflow and the randomisation procedure. Written informed consent will be obtained from patients in the first instance. If the patient is assessed and deemed not have the capacity to provide written informed consent then their legally authorised representative and/or caregivers will be approached to give agreement on behalf of the patient according to GCP.

During the screening visit (between 1 week and 1 day before operation), the inclusion and exclusion criteria will be assessed. If all criteria are fulfilled and informed consent is given, the patient will be included as candidates. Trained members of the research team will enrol participants and implement the assignment of participants to either anaesthesia groups according to the centre randomisation the day before surgery. Group assignment

 Li T, *et al. BMJ Open* 2017;**7**:e016937. doi:10.1136/bmjopen-2017-016937

will be revealed to members of the anaesthetic team only when the patient enters the operating room.

### Preanaesthetic assessment

After a patient has given informed consent, preanaesthetic assessment will be conducted including demographics, medical history, physical examination, cognitive status, comorbidities, current medication use and assessment of preoperative quality of life. Then a specific form will be documented within 24 hours before surgery.

Data collectors will visit recruited patients from preanaesthetic assessment the day before surgery to 7 days after surgery, discharge and follow-up patients at 6 and 12 months after the surgery to collect required data. Data collectors will receive specific training of using of CAM, DRS-R-98, MMSE and other tests used in this trial and will be blinded to the group allocation.

Premedication for anaesthesia will not be encouraged before surgery. Any medication impairing cognitive function will be avoided.

### Anaesthesia procedure

GA group: For patients assigned to receive GA, anaesthesia will be induced with propofol (or etomidate), sufentanil (or fentanyl) and will be maintained with either intravenous (propofol), inhalational (sevoflurane or isoflurane with or without nitrous oxide) through laryngeal mask airway, tracheal intubation, mechanical ventilation or combined intravenous–inhalational anaesthetics. Iliac fascia '3-in-1', femoral nerve block or posterior lumbar plexus block, single nerve block or continuous nerve block are recommended blocking methods.

RA group: Regional anaesthetic techniques for the surgery include epidural anaesthesia, spinal anaesthesia and combined spinal and epidural, with or without additional peripheral nerve block according to anaesthetist experience. The type and dosage of local anaesthetic depend on anaesthetist experience as well. No sedative is administrated in RA group. Any medication impairing cognitive function, like midazolam or dexmedetomidine, will be avoided in both groups.

On the day of surgery, data collection will be performed and recorded with assessment of intraoperative and perioperative parameters as well as serious adverse events (SAEs). For perioperative information, data will be collected for duration and types of anaesthesia and surgery, doses of analgesics, sedatives and anaesthetic agents, intraoperative fluid balance, including blood or clotting products transfusion and postoperative analgesic consumption and use of drugs with anticholinergic properties.

### Postoperative observation and follow-up

Following surgery, patients will be sent to the postanaesthesia care unit (PACU) of each clinical centre for continuous routine vital signs monitoring, they will be discharged from PACU to the orthopaedic ward after they are assessed to have recovered from anaesthesia. Both groups will receive routine postoperative care on orthopaedic ward. Postoperative analgesia such as intravenous, epidural, nerve block analgesia and oral analgesics can be administered, according to the local procedures of each clinical trial site, aiming to maintain a VAS pain score ≤30 mm, but any postoperative sedatives will be avoided.

During the postoperative visits, the primary and secondary endpoints, adverse events (AEs) and SAEs will be assessed and documented, including CAM, DRS-98-R (if applicable), VAS, analgesic use (if applicable), sedative use (if applicable), postoperative morbidity and laboratory results, such as serum haemoglobin, haematocrit, leucocytes, aspartate aminotransferase, alanine aminotransferase, albumin, serum creatinine and urea concentrations, serum sodium and potassium and serum glucose concentration.

A daily delirium assessment will be performed for all randomised patients who will be followed up for 7 days after surgery or until discharge from the hospital. Over these 7 days, daily clinical assessments will be continued to evaluate duration of delirium until the symptoms of delirium are resolved or until the patient is discharged for patients who are delirious while daily assessments are discontinued for patients who are not delirious by day 7.

### Adverse events or serious adverse event

An AE[26] is described as any unpredictable or unfavourable clinical outcome associated with any medical interventions that occur during the study period. AE may be or may not be related to the study intervention. An SAE in the RAGA trial is defined as any unpredictable medical events that causes death, life threatening, results in significant incapacity/disability requires prolonged hospitalisation or is otherwise considered medically significant by the investigators.

In accordance with the guidance of GCP, AEs or SAEs will be reported using AE/SAE forms in patients Care Report Forms (CRF) including type, date, onset, severity, relation to intervention, management and outcome of these events. All AE/SAE will be monitored carefully, treated promptly if necessary based on clinical judgement and followed up until the events are properly resolved and patients are stabilised or recovered to normal.

### Concomitant treatment

The medication concomitant with the RA or GA should be as simple as possible. Previous study reports that several drugs may increase the risk of POD.[27–29] So the following medications such as intraoperative benzodiazepines are prohibited. Additionally, opioids (remifentanil, sufentanil, fentanyl or morphine) and muscle relaxant (rocuronium, atracurium or cisatracurium) will be administered when deemed clinically necessary by the attending anaesthesiologists or physicians. Dosage, route, unit frequency of administration, and indication for administration and dates of all concomitant medication should be captured and recorded in details if used in clinical necessity.

## Study withdrawal

A patient will be withdrawn from the trial for any of the following reasons. The reason for the participant being withdrawn from the trial will be recorded on the 'withdrawal/change of status' form:

1. Participants choose to withdraw consent;
2. If continuing, the trial is harmful to the patient's well-being;
3. The participant becomes unable to complete the trial documentation and the trial investigators feel it is no longer appropriate for the participant to continue;
4. SAEs that are deemed related to the trial interventions;
5. Safety reasons determined by the trial admission or the advisory board.

## Data management

The data will be managed and analysed according to the guideline of GCP and handled in strictest confidence. All trial data will be recorded into an electronical case report form (eCRF) by investigator-designated and appropriately trained personnel. The eCRF must be completed as soon as possible after the source document information is collected. An explanation must be given for any missing data. A copy will be retained at the trial centre, and the original eCRF will be sent via web-based submitting system to trial coordinating centre after accuracy and completeness checking by the data monitoring staff.

The completed CRF must be reviewed and input into a password protected trial database with double independent entries. After checking for plausibility, consistency and completeness, the database will be exported into the Statistical Analysis System (SAS). Appropriate backup copies of the database and related software files will be maintained. Databases are backed up by the database administrator in conjunction with any updates or changes to the database.

## Data sharing statement

The data gathered will be anonymised for processing. Data will be held securely and used only by the researcher and supervisor. It will be held for a maximum of 3 years to inform the project and will then archived in an anonymised form and deposited.

All data generated during the project will be made freely available via the Research Data Repository of The Second Affiliated Hospital and Yuying Children's Hospital of Wenzhou Medical University. DOIs to these data will be provided and cited in any published articles using the data and any other data generated in the project. No further security, licensing or ethical issues are related to the expected data. Any data relevant to a published article will be made available alongside the article when published.

## Statistics

### Sample size estimation

The sample size calculation is based on incidence rate of POD. Previous published incidence of POD diagnosed using CAM ranged from 28% to 50% in the elderly patients (age 65) during hospital stay for hip fracture surgery. Observational studies in China estimated 3-day POD incidences of 11.1%–23.3%.[30–32] Assuming that the 7-day GA Group in the present study will have a similar average delirium incidence as in previous 3-day studies, we adopted a conservative estimation of 7-day postsurgery POD incidence undergoing GA of 26% and a reduction of 30% POD in group RA. Then 441 patients should be needed for each group to give power ($1-\beta$) of 0.80 and the significance level of 0.05 (two sided). Losses to follow-up for the primary outcome are estimated to be 10.0%. Thus, a total of 1000 patients will be recruited from the nine clinical trial centres in China.

## Statistical analysis

Statistical analyses will be performed with SAS (SAS Institute) by the statistician team of the trial. Prior to the analysis of the final study data, a detailed Statistical Analysis Plan will be developed to describe all analyses that will be performed. Data will be primarily analysed according to the intention-to-treat (ITT) principle and additional evaluation per protocol will be compared with those of the ITT analysis.

All tests will be two sided and statistical significance will be considered at $p<0.05$. All parameters will be calculated and presented with estimates and 95% CIs. Differences between groups will be calculated. Primary outcome (the 7-day incidence of POD) will be analysed using $\chi^2$ tests. Continuous secondary outcome measures, if normally distributed, will be tested using Student's t-test, otherwise Mann-Whitney test will be employed. Categorical outcome variables will be tested with a $\chi^2$ test. The time-to-event data, such as delirium-free duration or discharge from hospital will be tested using a log-rank test. Safety and tolerability data will be summarised by treatment groups and will include the number of patients, the rate of occurrence, the severity and relationship to interventions.

Possible confounding factors include demography, fracture characteristics, type of surgery and perioperative and postoperative complications. Stratification will act as a fundamental adjustment method for controlling the most essential confounders. Subgroup analysis according to variables relevant to the risk of POD (age, existing cognitive impairment, preoperative delirium) will help to define the risks of POD. Baseline measurements of the outcome variables, together with factors such as age, gender, body mass index, comorbidities and patient preference, will be included as covariates. Model-adjusted ORs between the two treatment groups will be estimated and tested with logistic regression for adjusting baseline measures to compare differences between groups in the primary and in the secondary outcome measures.

## Ethical and management considerations

The trial will be conducted in compliance with the protocol and GCP. Monitoring will be predefined in a study-specific monitoring manual, and will be conducted by Clinical Research Unit of The Second Affiliated Hospital and Yuying Children's Hospital of Wenzhou

Medical University according to approved standard operating procedures. The monitor will verify the enrolment of patients, check the informed consent forms, verify the source data and entries into the CRF and regularly supervise the progress of the trial to assist the local investigators in conducting the study according to the protocol, as well as to meet regulatory and ethical requirements. In accordance with the standard operating procedures and policies of the local institutional review board (IRB)/independent ethics committee, the site investigator will report SAEs to the PI and the IRB of the Second Affiliated Hospital and Yuying Children's Hospital of Wenzhou Medical University.

### Trial status

The current protocol is version 1.1. The randomised trial, which commenced in September 2014, is currently in the phase of participant enrolment and follow-up. To date (31 August 2016), 968 patients have been screened and 399 patients have been randomised in this study. Recruitment of patients is about 50% slower than expected, so the recruitment period of this trial will be extended from January 2016 to July 2018.

## DISCUSSION

The RAGA-delirium trial is to allow us to detect whether RA given to older patients undergoing hip fracture surgery is related to a significantly lower incidence of POD. Hip fracture will present a real social and economic challenge as the population ages and the number of fractures increases. Both RA and GA are widely used in hip fracture surgery. Currently available evidence is controversial as to whether GA accelerates the cognitive impairment or whether RA reduces POD. A study examining elective surgery suggests no difference in POD when RA and GA are compared.[33] However, the results require verification as the study was underpowered (n=29 per group).[34] A recently published result of a clinical trial shows a contradictive conclusion that the 30-day mortality was marginally reduced for spinal anaesthesia 7/164 (4.3%) vs 5/158 (3.2%) (p=0.57), while at 1 year it was lower for GA 20/163 (12.1%) vs 32/158 (20.2%) (p=0.05). Methodological rigour in particular regarding randomisation allocation concealment and assessor blinding of these studies was suboptimal.[5] The numbers of participants included are insufficient to eliminate a difference between the two techniques in the majority of outcomes studied. Large randomised trials reflecting actual clinical practice are required before drawing final conclusions.

### Strengths and limitations

In the present trial, a more robust methodology is adopted to decrease risk of potential bias. First, we are one of the few academic clinical research teams equipped with web-based central randomisation in China.

Usage of computer-generated assignment to assign randomisation to recruited patients to each group;

additionally covariate adaptive randomisation is also possible using the method of minimisation to assess the imbalance of sample size among several covariates.

Clinical trials display greater difficulty instituting 'blinding', we require data collectors and outcome assessors to remain blinded to group allocation in order to prevent subsequent differential cointerventions or biased assessment of outcomes to minimise performance bias and a Hawthorne effect.[23] These measures ensure grouping balance against bias in all the respects except the intervention each group received.[35]

Other strength is the pragmatic and multicentre design. We adopted a pragmatic design with the aim of depicting the outcomes of anaesthesia methods in real-life routine clinical practice which are more representable of clinical practice and producing results that can be generalised. Therefore, the anaesthesia methods for both groups will not be restricted to one or two specific methods in the trial.

On the other hand, however, the pragmatic and multicentre design may limit the accuracy of measurement of interventions and the plausibility of comparison. Unlike randomised clinical trial for medications, there is variation in interventions performed by anaesthetists in different centres due to the operational characteristics of anaesthesia. Furthermore, there is variability in the surgical interventions performed in various centres due to surgeon variability and numerous other variables. This compromise of sample and result homogeneity could introduce biases influencing both arms, but their effect will be diluted with large sample size.

## CONCLUSION

In conclusion, this study will provide clinical evidence on the safety of RA and GA in older patients for hip fracture surgery. We propose to contribute to the emerging literature by developing a more robust pragmatic multicentre randomised controlled clinical trial. The results of the trial are expected to help anaesthetists in selecting appropriate anaesthesia in a specific subgroup of patients and especially those at high risk for POD.

**Author affiliations**
[1]Department of Anesthesia, The Second Affiliated Hospital and Yuying Children's Hospital of Wenzhou Medical University, Wenzhou, Zhejiang, China
[2]Perioperative, Critical Care and Trauma Trials Group, College of Medical and Dental Sciences, University of Birmingham, Birmingham, UK
[3]Academic Department of Anaesthesia, Critical Care, Pain and Resuscitation, Birmingham Heartlands Hospital, Heart of England NHS Foundation Trust, Birmingham, UK
[4]Key Lab. of Reproduction Regulation of NPFPC, SIPPR, IRD, Fudan University, Shanghai, China

**Contributors** All authors have made substantial contributions to the conception or design of the work, or the acquisition, analysis or interpretation of data. TL and YZ drafted the manuscript. JY, JL, YG, TM, YW, QL and FG revised it critically for important intellectual content. TM edited the language. All authors approved final version and agreed to be accountable for all aspects of the work in ensuring that questions related to the accuracy or integrity of any part of the work are appropriately investigated and resolved.

**Funding** This work was funded by Zhejiang Health and Family Planning Commission Programme and the Recruitment Program of Global Experts (Grant number: 2014PYA015), China.

**Competing interests** None declared.

**Patient consent** Obtained.

**Ethics approval** The study has been approved by the IRB of The Second Affiliated Hospital and Yuying Children's Hospital of Wenzhou Medical University and by the local IRB of the participating centres.

**Provenance and peer review** Not commissioned; externally peer reviewed.

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
