## [Reviewer comments · BMJ Open]

ARTICLE DETAILS

TITLE (PROVISIONAL)	Comparison of regional with general anaesthesia on post-operative delirium (RAGA-delirium) in the older patients undergoing hip fracture surgery: study protocol for a multicentre randomized controlled trial
AUTHORS	Li, Ting; Yeung, Joyce; Li, Jun; Zhang, Yan; Melody, Teresa; Gao, Ye; Wang, Yi; Lian, Qian-Quan; Gao, Fang

VERSION 1 - REVIEW

REVIEWER	Michael Gillies Royal Infirmary of Edinburgh University of Edinburgh Edinburgh UK
REVIEW RETURNED	08-May-2017

GENERAL COMMENTS	The authors are to be commended for undertaking this important and well designed study. I think this manuscript can be accepted with only very minor adjustments and I have no major concerns. Minor comments: ABSTRACT pp2 Line 48 Sentence beginning "At the rapidly changing era of China..." should be rewritten METHODS AND DESIGN pp6 Line 20-37. Would the contraindication to neuraxial blockade include severe valvular heart disease or other heart disease? pp9 Line 24-37: Please could the authors explain the rationale to allow premedication as these drugs could alter the incidence of delirium. If premedication was permitted could or should it be standardised? pp14 Line 30-32 It would read better if the sentences, "Recruitment of patients..." and " So the recruitment..." were joined with a comma. DISCUSSION pp14 Line 44 Add "as to" between the words "controversial" and "whether" pp14 Line 56 "was" instead of "were"
---

REVIEWER	Coburn Mark Department of Anesthesiology University Hospital RWTH Aachen Germany
REVIEW RETURNED	12-May-2017

GENERAL COMMENTS	Li Ting and co-workers submitted a study protocol “Comparison of regional with general anaesthesia on post - operative delirium (RAGA-delirium) in the older patients undergoing hip fracture surgery: study protocol for a multicentre randomized controlled trial”. Indeed there is a need for large scale rct assessing effectiveness of regional vs general anesthesia in regard to postoperative delirium. However there are several major concerns about the study protocol. This is of importance as the study has already recruited more than 400 patients and recruitment is estimated to be finished by September 2017. I do not see the benefit of the study protocol being published at this stage of the study. Introduction:  • Postoperative delirium is a clearly defined and established term. There is no need to try to rewrite delirium. • The authors should refer to the systematic review and meta-analysis which assesses the association of postoperative delirium in hip fracture patients (Sanders RD et al. BMJ 2011; 343: d4331 • Further they could refer to a recent editorial to underline their goals (Coburn M et al. EJA 2017; 34: 115-117) • It is not clear what authors mean by regional anaesthesia, this needs to be clarified throughout the text Methods:  • In this pragmatic approach authors are combining nerve blocks with GA and compare that with SA or Epidural anesthesia. This is not described, see comment above. Further it is not clear if patients are able to receive sedation with “regional anesthesia” and how this can be provided. • There is cumulating evidence that avoiding very low BIS values are associated with increased POD rates, see latest guideline (ESA guideline on POD. EJA 2017; 34: 192-214) • The authors are calculating a drop-out rate of 10% for the primary outcome parameter after randomization. This is critical, what are the reasons why you expect a drop-out after randomization? A PP and ITT analysis should be performed. • At which time intervals is POD assessed, twice a day, fixed time-points? • How do authors perform a MMSE testing at long-term follow-up? This test cannot be carried out by telephone? • The manuscript needs language editing
---

VERSION 1 – AUTHOR RESPONSE

Reviewer: 1

ABSTRACT

1. Comment: pp2 Line 48 Sentence beginning "At the rapidly changing era of China..." should be rewritten

Response: Thanks for the reviewer’s kind suggestion. The sentence has been rewritten to “At the era of aging and increasing emphasis on delirium prevention in China, this trial has the potential to inform

the future national guideline to reduce POD.”(please see the text in the manuscript).

METHODS AND DESIGN

2. Comment: pp6 Line 20-37. Would the contraindication to neuraxial blockade include severe valvular heart disease or other heart disease?

Response: Thank you for the comment. Patients with severe systemic disease such as Severe valvular heart disease or other heart disease, though are not absolute contraindication to neuraxial blockade. The anaesthetist will decide before randomization according to the guideline of the site.

3. Comment: pp9 Line 24-37: Please could the authors explain the rationale to allow premedication as these drugs could alter the incidence of delirium. If premedication was permitted could or should it be standardised?

Response: Thank you for the comment. Premedication for anaesthesia will not be encouraged in the protocol. Any medication impairing cognitive function, like midazolam or dexmedetomidine, will be avoided. If there is any premedication administered when deemed clinically necessary, it must be recorded in detail.

4. Comment: pp14 Line 30-32 It would read better if the sentences, "Recruitment of patients..." and "So the recruitment..." were joined with a comma.

Response: Thank you for the kind reminder. We have revised it in the manuscript text.

DISCUSSION

5. Comment: pp14 Line 44 : Add "as to" between the words "controversial" and "whether"

Response: Thank you for the kind reminder. We have revised it in the manuscript text.

6. Comment: pp14 Line 56 : "was" instead of "were"

Response: Thank you for the kind reminder. We have revised it in the manuscript text.

Reviewer: 2

7. Comment: Li Ting and co-workers submitted a study protocol “Comparison of regional with general anaesthesia on post - operative delirium (RAGA-delirium) in the older patients undergoing hip fracture surgery: study protocol for a multicentre randomized controlled trial”. Indeed there is a need for large scale rct assessing effectiveness of regional vs general anesthesia in regard to postoperative delirium. However there are several major concerns about the study protocol. This is of importance as the study has already recruited more than 400 patients and recruitment is estimated to be finished by September 2017. I do not see the benefit of the study protocol being published at this stage of the study.

Response: Thank you for the comment. We are trying to publish our study protocol because we are doing our best to verify that the future research report will adhere to the original plan for trial conduct and analysis. Currently the trial is in the middle of recruitment. The estimated recruitment will be finished on July 2018.

Introduction:

8. Comment: Postoperative delirium is a clearly defined and established term. There is no need to try to rewrite delirium.

Response: Thank you for your comment. The sentence “Delirium, sometimes called “acute confusional state”, is a common clinical syndrome characterised by disturbed consciousness and a change in cognitive function or perception which develops over a short period of time (usually 1-2 days).” have been deleted.

9. Comment: The authors should refer to the systematic review and meta-analysis which assesses the association of postoperative delirium in hip fracture patients (Sanders RD et al. BMJ 2011; 343: d4331

Response: Thank you for the kind instruction. the suggested review clearly illustrated the Anticipating and managing postoperative delirium and cognitive decline in adults. We have referred to it in the background chapter of the manuscript.

10. Comment: Further they could refer to a recent editorial to underline their goals (Coburn M et al. EJA 2017; 34: 115-117)

Response: Thank you for the thoughtful instruction. The editorial clearly told the never-ending story of the elderly with fractured neck of femur. We have got support for our trial and have referred to it in the background chapter of the manuscript.

11. Comment: It is not clear what authors mean by reginal anaesthesia, this needs to be clarified throughout the text

Response: Thank you for the comment. Regional Anaesthesia include epidural anaesthesia, spinal anaesthesia, and combined spinal and epidural, with or without additional peripheral nerve block (see pp9 42-47).

12. Comment: In this pragmatic approach authors are combining nerve blocks with GA and compare that with SA or Epidural anesthesia. This is not described, see comment above. Further it is not clear if patients are able to receive sedation with “regional anestehsia” and how this can be provided.

Response: Thank you for the comment. No sedative is administrated in RA group. Any medication impairing cognitive function, like midazolam or dexmedetomidine, will be avoided in both groups. We have added the sentences into the text.

13. Comment: There is cumulating evidence that avoiding very low BIS values are associated with increased POD rates, see latest guideline (ESA guideline on POD. EJA 2017; 34: 192-214)

Response: Thank you for the suggestion. BIS values can help controlling the depth of GA, but it is not a general practice to equip a BIS machine Sin many sites. So we are not able to collect BIS values in this trial.

14. Comment: The authors are calculating a drop-out rate of 10% for the primary outcome parameter after randomization. This is critical, what are the reasons why you expect a drop-out after randomization? A PP and ITT analysis should be performed.

Response: Thank you for the comments.

A. The sample size calculation has been taken into account of available data. The dropout rate of follow-up among patients undergoing hip fracture surgery was less than 4% based on pilot data from one study site, that is why we made an estimate of 10% of dropout and a total of 1000 patients will be recruited during the trial.

B. An intention-to-treat (ITT) approach has been showed in the statistical chapter of the manuscript at pp13 line 9-11” Data will be primarily analysed according to the intention-to-treat (ITT) principle and additional evaluation per protocol (PP) will be compared with those of the ITT analysis.” .

15. Comment: At which time intervals is POD assessed, twice a day, fixed time-points?

Response: Thank you for your comment. POD is assessed once a day(see pp10 32-33)

16. Comment: How do authors perform a MMSE testing at long-term follow-up? This test cannot be carried out by telephone?

Response: Thank you for the comment. We perform the MMSE by Wechat (a very popular social media with over seven billion active users in China). WeChat is a cross-platform social media in China with over a billion active users. Most of hospitals in China have their own public accounts of Wechat to communicate with the public. Most adults and quite many of old people using Wechat for communication. In this trial, we can collect the MMSE data with the patients or their carers using Wechat when they go home.

17. Comment: The manuscript needs language editing

Response: some of the authors are native British. Teresa Melody, who is a born native British and an experienced medical staff and researcher, has edited the English language.

VERSION 2 – REVIEW

REVIEWER	Michael Gillies Royal Infirmary of Edinburgh, Edinburgh Uk
REVIEW RETURNED	30-Jun-2017

GENERAL COMMENTS	I am satisfied with the revision and have no further comments.
--